# PP19128R, a Multiepitope Vaccine Designed to Prevent Latent Tuberculosis Infection, Induced Immune Responses *In Silico* and *In Vitro* Assays

**DOI:** 10.3390/vaccines11040856

**Published:** 2023-04-17

**Authors:** Fan Jiang, Cong Peng, Peng Cheng, Jie Wang, Jianqi Lian, Wenping Gong

**Affiliations:** 1Tuberculosis Prevention and Control Key Laboratory/Beijing Key Laboratory of New Techniques of Tuberculosis Diagnosis and Treatment, Senior Department of Tuberculosis, The 8th Medical Center of PLA General Hospital, Beijing 100091, China; 2The Second Brigade of Cadet, Basic Medical Science Academy of Air Force Medical University, Xi’an 710032, China; 3Department of Infectious Diseases, Tangdu Hospital, Air Force Medical University, Xi’an 710032, China; lianjq@fmmu.edu.cn

**Keywords:** tuberculosis (TB), *Mycobacterium tuberculosis* (MTB), multiepitope vaccine (MEV), immunoinformatics, cytometric bead assay (CBA), enzyme-linked immunospot (ELISpot) assay

## Abstract

**Background:** Latent tuberculosis infection (LTBI) is the primary source of active tuberculosis (ATB), but a preventive vaccine against LTBI is lacking. **Methods:** In this study, dominant helper T lymphocyte (HTL), cytotoxic T lymphocyte (CTL), and B-cell epitopes were identified from nine antigens related to LTBI and regions of difference (RDs). These epitopes were used to construct a novel multiepitope vaccine (MEV) based on their antigenicity, immunogenicity, sensitization, and toxicity. The immunological characteristics of the MEV were analyzed with immunoinformatics technology and verified by enzyme-linked immunospot assay and Th1/Th2/Th17 cytokine assay in vitro. **Results:** A novel MEV, designated PP19128R, containing 19 HTL epitopes, 12 CTL epitopes, 8 B-cell epitopes, toll-like receptor (TLR) agonists, and helper peptides, was successfully constructed. Bioinformatics analysis showed that the antigenicity, immunogenicity, and solubility of PP19128R were 0.8067, 9.29811, and 0.900675, respectively. The global population coverage of PP19128R in HLA class I and II alleles reached 82.24% and 93.71%, respectively. The binding energies of the PP19128R-TLR2 and PP19128R-TLR4 complexes were −1324.77 kcal/mol and −1278 kcal/mol, respectively. In vitro experiments showed that the PP19128R vaccine significantly increased the number of interferon gamma-positive (IFN-γ^+^) T lymphocytes and the levels of cytokines, such as IFN-γ, tumor necrosis factor-α (TNF-α), interleukin-6 (IL-6), and IL-10. Furthermore, positive correlations were observed between PP19128R-specific cytokines in ATB patients and individuals with LTBI. **Conclusions:** The PP19128R vaccine is a promising MEV with excellent antigenicity and immunogenicity and no toxicity or sensitization that can induce robust immune responses in silico and in vitro. This study provides a vaccine candidate for the prevention of LTBI in the future.

## 1. Introduction

Tuberculosis (TB) is an infectious disease caused by *Mycobacterium tuberculosis* (MTB) infection. Bacillus Calmette–Guérin (BCG) is the only widely used vaccine for the prevention of TB, but BCG has shown high heterogeneity in the efficacy of protection against adult TB [1,2]. In 2021, there were 10.4 million new TB cases and 1.4 million deaths worldwide [3]. The World Health Organization (WHO) has launched a series of programs to implement a 20 year plan to end TB, aiming for a 95% reduction in TB deaths and a 90% reduction in TB infections by 2035 (based on 2015 data) and to eliminate TB by 2050 [3]. The development and production of vaccines are, therefore, essential to achieving this goal.

BCG was first administered to a newborn in Paris in 1921. Over the past century, BCG has remained the only licensed vaccine for the prevention of human TB, saving millions of lives and immunizing more people than any other vaccine [4,5]. However, the protective efficacy of BCG is highly variable, ranging from 0% to 80%, and the protection lasts only 10–20 years [4,6,7,8,9]. Fortunately, new TB vaccines have been actively developed over past decades, such as inactivated, live attenuated, subunit, and viral vector-based TB vaccines [10,11]. However, the promising subunit vaccine, M72/AS01E, showed only 49.7% (95% CI 2.1–74.2) efficacy in the final analysis in a clinical trial [12].

Latent tuberculosis infection (LTBI) is “a state of persistent immune response to stimulation by MTB antigens with no evidence of clinically manifest active TB” [13] and “refers to a host who is TB immunoreactive in the absence of TB disease” [14]. A previous study reported that the global prevalence of LTBI was 23.0% in 2014 [15], and individuals with LTBI have become a significant source of active tuberculosis (ATB) patients. However, there is no specific vaccine to prevent LTBI. Traditional vaccine development is an extremely lengthy process. With the advent of computer information technology, artificial intelligence, and machine learning [16,17], bioinformatics and immunoinformatics methods have been gradually developed [18,19], which has promoted the development of multiepitope vaccines (MEVs) based on reverse vaccinology into the fast track [20]. For example, the reagents and consumables needed for experiments to test allergic and autoimmune responses to vaccine molecules were prohibitively expensive in the past, whereas electronic analytical predictions are now available from an online server [20,21,22].

In this study, we aimed to develop a novel MEV for preventing LTBI based on in silico analysis and in vitro experiments. First, we analyzed the antigenicity, immunogenicity, physicochemical properties, secondary structure, and tertiary structure of the MEV and its immune responses. Second, an MEV was successfully constructed and expressed in *Escherichia coli* (*E. coli*). Third, the number of MEV-induced IFN-γ^+^ T lymphocytes was detected using the enzyme-linked immunospot (ELISpot) assay. In addition, the levels of Th1/Th2/Th17 cytokines produced by peripheral blood mononuclear cells (PBMCs) collected from healthy subjects, ATB patients, and individuals with LTBI were determined using the cytometric bead assay (CBA) technique. Finally, potential correlations between individual cytokines in the three populations were analyzed. This study provides a new potential vaccine candidate for LTBI prevention and highlights a method to evaluate the agreement between in silico analysis and in vitro experiments for MEV.

## 2. Materials and Methods

### 2.1. In Silico Analysis

#### 2.1.1. Selection of Candidate Antigens for MTB

Based on bibliometric and bioinformatics techniques, we identified 21 potential antigens derived from latent tuberculosis infection and regions of difference (LTBI-RD) [23,24,25]. These latent tuberculosis infection and region of difference (LTBI-RD) antigens were considered the most promising candidates for diagnosing and preventing LTBI. In the current study, 17 LTBI-RD antigens (Rv1511, Rv1736c, Rv1737c, Rv1980c, Rv1981c, Rv2031c, Rv2626c, Rv2653c, Rv2656c, Rv2659c, Rv2660c, Rv3425, Rv3429, Rv3872, Rv3873, Rv3878, and Rv3879) were selected to identify immunodominant epitopes.

#### 2.1.2. Screening for Immunodominant T Cell Epitopes

Helper T lymphocyte (HTL) and cytotoxic T lymphocyte (CTL) epitopes were predicted from the selected LTBI-RD antigens using the MHC II server (http://tools.iedb.org/mhcii/ (accessed on 4 February 2022) and MHC I server (http://tools.iedb.org/mhci/ (accessed on 4 February 2022) in the IEDB database, respectively, following our previous studies [8,23,24,26,27]. HTL and CTL epitopes with percentile ranks < 0.5 were considered immunodominant. VaxiJen v2.0 (http://www.ddg-pharmfac.net/vaxijen/VaxiJen/VaxiJen.html (accessed on 4 February 2022) was used to predict the antigenicity of HTL, and CTL epitopes. Then, HTL and CTL epitopes with antigenicity scores > 0.7 were identified for further analysis following a previous study [28]. Furthermore, AllerTOP v.2.0 (http://www.ddg-pharmfac.net/AllerTOP/ (accessed on 7 February 2022) and Allergen FP v.1.0 (http://ddg-pharmfac.net/AllergenFP/) were used to predict the allergenicity of HTL and CTL epitopes in accordance with previous studies [29,30]. In addition, the toxicity of HTL and CTL epitopes was predicted using Toxin Pred (http://crdd.osdd.net/raghava/toxinpred/ (accessed on 7 February 2022), the interferon gamma (IFN-γ) inducibility of HTL epitopes was analyzed using the IFN Epitope Server (http://crdd.osdd.net/raghava/ifnepitope/index.php (accessed on 7 February 2022), and the immunogenicity of CTL epitopes was determined using the Class I Immunogenicity Server (http://tools.iedb.org/immunogenicity/ (accessed on 7 February 2022). Finally, epitopes with an immune score > 0 (a higher score could induce a more robust immune response) were used for the subsequent analysis based on our previous extensive bioinformatics discoveries [23,31].

#### 2.1.3. Immunodominant B-Cell Epitope Prediction

Linear B-cell epitopes of LTBI-RD antigens were predicted using the ABC pred server (https://webs.iiitd.edu.in/raghava/abcpred/) following a previous study [32]. The server identifies linear B-cell epitopes based on trained recurrent neural network scores. Higher epitope scores indicate their ability to induce a more robust immune response.

#### 2.1.4. Analysis of Population Coverage of HLA-I and HLA-II Alleles and Construction of MEV

To develop a novel MEV, the immunodominant HTL, CTL, and B-cell epitopes were screened and identified based on their adjusted rank, antigenicity, immunogenicity, toxicity, and sensitization. Furthermore, the population coverage of HLA-I and HLA-II alleles, which restrict HTL and CTL immunodominant epitopes, was determined using the IEDB database (http://tools.iedb.org/population/ (accessed on 10 February 2022). Genotypic frequencies of HLA-I and HLA -II alleles were selected from the Allele Frequency database (http://www.allelefrequencies.net/).

A novel MEV was constructed and named PP19128R. Linkers such as GPGPG, AAY, and KK were used to link HTL, CTL, and B-cell epitopes. Toll-like receptor 2 (TLR2) agonist Porin B (PorB) [33] (IALTLAALPVAAMADVTLYGTIKAGVETSRSVAHNGAQAASVETGTGIVDLG-SKIGFKGQEDLGNGLKAIWQVEQ) and TLR-4 agonist RS-09 [34] (APPHALS) were used as adjuvants to enhance the immunogenicity of MEV. A pan HLA DR binding epitope (PADRE) (AGLFQRHGEGTKATVGEPV) was added to induce a more robust HTL response [35]. Finally, a 6-His tag was inserted at the end of the amino acid sequence to purify the PP19128R vaccine.

In addition, the antigenicity, immunogenicity, allergenicity, sensitization, and toxicity of the PP19128R vaccine were analyzed using VaxiJen v2.0 [28], ANTIGENpro [36], IEDB Immunogenicity server [37], AllerTOP v.2.0 [29], Allergen FP v.1.0 [30], and Toxin Pred server [38].

#### 2.1.5. Prediction of Physicochemical Properties and Secondary/Tertiary Structure of the PP19128 Vaccine

Based on previous studies, the Expasy Protparam server (https://web.expasy.org/protparam/ (accessed on 13 February 2022), Scratch Protein Predictor (http://scratch.proteomics.ics.uci.edu/ (accessed on 13 February 2022), PSIPRED server (http://bioinf.cs.ucl.ac.uk/psipred/ (accessed on 13 February 2022), and RaptorX Property (http://raptorx.uchicago.edu/StructurePropertyPred/predict/ (accessed on 13 February 2022) were used to predict the physicochemical parameters, solubility, and secondary structure of PP19128R vaccine, respectively [39,40,41,42].

Using the multi-threaded method LOMETS, templates for the space structure of the PP19128R vaccine were automatically extracted using the I-TASSER server (https://zhanggroup.org//I-TASSER/ (accessed on 16 February 2022). Furthermore, the best model with the highest C-score was selected for optimization using the GalaxyRefine web server (https://galaxy.seoklab.org/cgi-bin/submit.cgi?type=REFINE (accessed on 16 February 2022). Verification of the optimized structure model was confirmed using the ProSA web server (https://prosa.services.came.sbg.ac.at/prosa.php (accessed on 16 February 2022) and the ERRAT server (https://saves.mbi.ucla.edu/ (accessed on 16 February 2022) following our previous studies [8,23,26,27]. Potential protein structure errors were detected using the ProSA web server based on the Z-score, and a Z-score > 0 indicates that there may be an error or unstable part in the protein model. Ramachandran plots were generated using UCLA-DOE LAB—SAVES v6.0 (https://saves.mbi.ucla.edu/ (accessed on 16 February 2022) following previous studies [43,44,45,46].

#### 2.1.6. Molecular Docking between the PP19128 Vaccine and TLRs and Their Simulated Immune Responses

The Protein Data Bank (PDB) files for TLR2 and TLR4 were downloaded from the Molecular Modeling Database (MMDB, https://www.ncbi.nlm.nih.gov/structure/ (accessed on 18 February 2022). Then, molecular docking between the PP19128R vaccine and TLR2 or TLR4 was analyzed using the ClusPro2.0 server (https://cluspro.bu.edu/home.php (accessed on 18 February 2022) following a previous study [47]. Finally, the simulation of immune responses induced by the PP19128R vaccine was predicted by using the C-ImmSim server (https://150.146.2.1/C-IMMSIM/index.php (accessed on 18 February 2022) [48].

### 2.2. In Vitro Experiments

#### 2.2.1. Ethics and Experimental Subjects

This study was conducted between April 2022 and December 2022 at the Senior Department of Tuberculosis, Eighth Medical Center of PLA General Hospital. The experiments on blood samples collected from health controls (HCs), ATB patients, and individuals with LTBI were approved by the Ethics Committee of the Eighth Medical Center of PLA General Hospital (approval number 309202204080808). In addition, all subjects signed an informed consent form.

The inclusion and exclusion criteria for HCs, ATB patients, and individuals with LTBI were determined according to the tuberculosis diagnostic criteria (WS288-2017) formulated by the National Health and Family Planning Commission of China (NHFPC). The inclusion criteria for HCs were: (1) no contact history with ATB patients, (2) interferon gamma release assay (IGRA) or CE (fusion protein of culture filtrate protein 10 and 6 kDa early secretory antigenic target) test was negative, (3) no clinical manifestations of ATB, (4) human immunodeficiency virus (HIV)-negative, and (5) normal chest X-ray excluded the diagnosis of ATB. The exclusion criteria for HCs were (1) a history of travel or residence in high-risk TB areas, (2) TB hospital or laboratory staff, (3) children under 12 years of age, (4) patients with a history of TB or old lesions on lung imaging, (5) unable to perform CE antigen test or allergic, and (6) HIV-positive.

As an essential exploratory study, the ATB patients included in this study were characterized by MTB infection in the lung tissue, trachea, bronchus, and pleura, and the diagnosis of ATB was based on the Health Industry Standard of the People’s Republic of China: “Diagnosis of Tuberculosis WS 288-2017”. Exclusion criteria for ATB were (1) hormone users; (2) diseases affecting immune function, such as HIV infection, post-transplantation, and autoimmune diseases; (3) children under 12 years of age; and (4) malnourished individuals.

Inclusion criteria for individuals with LTBI were (1) a history of close contact with ATB patients, (2) staff of a specialized TB hospital or laboratory, (3) no clinical manifestations of ATB, (4) normal chest radiographs, (5) IGRA-positive, (6) HIV-negative, and (7) over 12 years of age. In addition, exclusion criteria for persons with LTBI were (1) ATB patients, (2) pregnant or lactating women, (3) HIV-positive, (4) anti-TB treatment for more than one month, and (5) children under 12 years of age.

#### 2.2.2. Construction of Recombinant Plasmid and Expression of the PP19128R Vaccine In Vitro

The nucleotide sequence of the PP19128R vaccine was artificially synthesized in vitro and then inserted into the pET28a(+) plasmid through BamHI and XhoI restriction sites to construct a recombinant pET28a(+)-PP19128R plasmid. The pET28a(+)-PP19128R recombinant plasmid was then transformed into *E. coli* BL21(DE3) for in vitro expression. The transformed *E. coli* strain was grown on Luria-Bertani (LB) solid plates (100 µg/mL kanamycin) overnight at 37 °C. First, individual colonies were picked and inoculated into liquid LB medium (100 µg/mL kanamycin, 5 ml, 37 °C, 220 rpm) and cultured overnight at 37 °C. Next, the first passage strains (1 ml) were added to liquid LB medium (100 µg/mL kanamycin, 100 mL, 37 °C, 220 rpm) and incubated for 4 to 6 h. Then, the first-generation bacteria were inoculated into a new 1 L volume of liquid LB medium (15 µg/mL kanamycin) at a 1% ratio and cultured at 37 °C, 220 rpm, until the optical density (OD) value of the bacterial solution reached 0.6–0.8. Then, 0.1 mM IPTG was added to the LB medium and incubated overnight at 16 °C, 220 rpm. The bacteria were centrifuged at 8000 rpm for 10 min to collect the cells and resuspended by adding the bacterial disruption solution (weight/volume = 1:15). After two bruises with a high-pressure homogenizer, the samples were centrifuged at 8000 rpm for 45 min at 4 °C, and the supernatant was collected. Finally, the PP19128R vaccine was purified through the C-terminal 6-His tag using Ni affinity chromatography and analyzed with sodium dodecyl sulfate-polyacrylamide gel electrophoresis (SDS-PAGE) following our previous studies [27,31].

#### 2.2.3. ELISpot

The numbers of IFN-γ+ T lymphocytes induced by the PP19128R vaccine and our previously developed TB vaccine HP13138PB [49] were determined using an ELISpot assay. First, a 5 mL blood sample was collected from HCs (*n* = 21), LTBI individuals (*n* = 25), and ATB patients (*n* = 19). PBMCs were separated from the blood sample using a human lymphocyte separation medium (Solarbio, Beijing, China, Cat: P8610) according to the manufacturer’s instructions. Then, 2.5 × 10^5^ PBMCs in 100 μL GIBCO AIM-V medium (Life Technology Invitrogen, California, USA, Cat. No. 087-0112DK) were added to one well of a 96-well ELISpot culture plate and incubated with 50 μL PP19128R or HP13138PB vaccine (100 μg/mL) or 50 μL AIM-V medium (negative control) in a CO_2_ incubator at 37 °C for 24 h. The number of spot-forming cells (SFCs) was detected using a Human IFN-γ ELISpot ^PRO^ kit (Mabtech AB, Nacka Strand, Sweden, Cat. No. 3420-2APW-10) according to the manufacturer’s instructions.

#### 2.2.4. Cytometric Bead Assay (CBA)

A 5 mL blood sample was collected from HCs (*n* = 7), LTBI individuals (*n* = 8), and ATB patients (*n* = 7). PBMCs were isolated from the blood samples as described above. Then, 2.5 × 10^5^ PBMCs in 100 μL AIM medium were added to one well of a 96-well cell culture plate (Mabtech AB, Nacka Strand, Sweden) and incubated with 50 μL PP19128R vaccine (100 μg/mL) in a CO_2_ incubator at 37 °C for 48 h. The mixture of cells and AIM medium was aspirated and then centrifuged at 1000 rpm for 10 min to collect the supernatant. The supernatant was gently transferred to another tube. The levels of interleukin-2 (IL-2), IL-4, IL-6, IL-10, IFN-γ, tumor necrosis factor-α (TNF-α), and IL-17A were measured using a BD CBA Human Th1/Th2/Th17 Cytokine Kit (BD Bioscience, San Diego, CA, USA, Cat: 560484) according to the manufacturer’s instructions. In addition, another group of HCs (*n* = 10) was enrolled as a negative control. Approximately 2.5 × 10^5^ PBMCs in 100 μL of AIM were added to a 96-well cell culture plate and incubated with 50 μL of AIM medium in a CO_2_ incubator at 37 °C for 48 h. Th1/Th2/Th17 cytokine levels were then detected as described above.

#### 2.2.5. Data Collation and Statistical Analysis

GraphPad Prism 9.5.0 software (San Diego, CA, USA) was used to analyze the data obtained from the ELISpot and CBA experiments. Briefly, the results of ELISpot and CBA experiments were analyzed by one-way ANOVA test or Kruskal–Wallis test, depending on normality and homogeneity of variance. In addition, principal component analysis (PCA) of cytokines was performed, and the correlations for the cytokines in each group were determined using the Pearson r method in GraphPad Prism 9.5.0 software. Simple linear regression was performed to analyze potential relationships between pairs of cytokines in each group. Data are presented as means ± the standard error of the mean (SEM), and *p* < 0.05 was considered a statistically significant difference.

## 3. Results

### 3.1. Identification of the Immunodominant Epitopes of HTL, CTL, and B-Cells and Analysis of the Population Coverage

Seventeen LTBI-RD antigens were used to identify the immunodominant HTL, CTL, and B-cell epitopes. Our results showed that 19 HTL, 12 CTL, and 8 B-cell epitopes were identified as immunodominant epitopes, and they were used to construct an MEV named PP19128R (Table 1). We then analyzed the population coverage of these immunodominant HTL and CTL epitopes using the IEDB database (Table 2). It was observed that the levels of population coverage for HLA-I alleles (which restrict CTL epitopes) in East Asia, South Asia, Southeast Asia, Southwest Asia, Northeast Asia, Central Africa, East Africa, North Africa, West Africa, South America, North America, Europe, Oceania, and the world were 85.63%, 70.59%, 91.15%, 64.29%, 92.53%, 57.38%, 61.10%, 65.52%, 59.42%, 65.56%, 82.17%, 82.00%, 76.33%, and 82.24%. Similarly, we found that the levels of population coverage for HLA-II alleles (which restrict HTL epitopes) in East Asia, South Asia, Southeast Asia, Southwest Asia, Northeast Asia, Central Africa, East Africa, North Africa, West Africa, South America, North America, Europe, Oceania, and the world were 73.85%, 97.75%, 88.86%, 83.80%, 97.97%, 82.75%, 89.22%, 80.01%, 88.35%, 98.12%, 99.93%, 99.06%, 97.55%, and 93.71%, respectively.

### 3.2. Construction of PP19128R Vaccine and Prediction of Its Physicochemical Properties and Secondary/Tertiary Structure

The results showed that the PP18128R vaccine contained 19 HTL, 12 CTL, and 8 B-cell immunodominant epitopes, and these epitopes were linked by GPGPG, AAY, and KK linkers, respectively. In addition, PorB, PADRE, and RS-09 were added to the PP18128R vaccine to improve the immunogenicity of the vaccine (Figure 1A). The PP18128R vaccine consisted of 930 amino acids and had a theoretical molecular weight of 98557.86 Da (Figure 1B). Secondary structure analysis of the PP19128R vaccine showed that it contained 39.46% alpha-helix, 11.61% extended strand, and 48.92% random coil (Figure 1C). In addition, the Expasy Protparam Server results showed that the PP19128R vaccine had a theoretical pI of 9.20; an estimated half-life of 20 h in mammalian reticulocytes, 30 min in yeast, and 10 h in *E. coli*; an instability index of 33.20; an aliphatic index of 79.32; a grand average of hydropathicity (GRAVY) of 0.04; and a solubility of 0.900675.

The ProSA web server and UCLA-DOE LAB SAVES v6.0 were used to validate 3D models of the PP19128R vaccine. As a result, the Z-scores of the PP19128R vaccine before and after optimization were −5.59 (Figure 2A) and −6.28 (Figure 2B), respectively. In addition, the Ramachandran plot suggested that the candidate model of the PP19128R vaccine contained 70.8% core, 22.8% allow, 4.4% gener, and 2.0% disall (Figure 2C). Interestingly, after optimization, these data changed to 87.2% core, 9.0% allow, 1.6% gener, and 2.2% disall (Figure 2D). In addition, the maximum deviation ratio of amino acid residues in the PP19128R vaccine was reduced from 23.2% to 18.8%.

In addition, five 3D models were generated by the I-TASSER server, and their C-scores were −1.39, −3.27, −3.98, −4.26, and −4.37 and generally between −2 and 5. The higher the score, the more accurate the model is. Therefore, model one (C-score = −1.39, estimated TM score = 0.54 ± 0.15, estimated RMSD = 12.1 ± 4.4 Å) was selected for further analysis (Figure 2E). Subsequently, the Galaxy Refine web server refined the vaccine models based on GDT-HA, RMSD, MolProbity, Clash score, Poor rotamers, and Rama favored score. Finally, model five, which had the highest GDT-HA values, lowest MolProbity values, and highest Rama favored score, was selected as the final 3D model for the PP19128R vaccine (Figure 2E).

### 3.3. The PP19128R Vaccine Binds Tightly to TLR2 and TLR4

The ClusPro2.0 server was used for molecular docking between the PP19128R vaccine and TLRs, generating 30 model complexes. The binding energies of these model complexes were analyzed, and the optimal PP19128R-TLR2 complex with the lowest binding energy was determined. It was found that the binding energy of the PP19128R-TLR2 complex was −1324.77kcal/mol (Figure 3A). Furthermore, we also explored the potential binding sites between the PP19128R and the TLR2, and there were 17 binding sites between the PP19128R vaccine and the TLR2 connected by a hydrogen bond (Figure 3B). Similarly, one of the models of the PP19128R-TLR4 complex with the lowest binding energy was used for further analysis. The binding energy of the PP19128R-TLR4 complex was −1278 kcal/mol (Figure 4A), and there were ten binding sites between the PP19128R vaccine and TLR4 connected by a hydrogen bond (Figure 4B).

### 3.4. The PP19128R Vaccine Induced a Robust Immune Response In Silico

Both specific and non-specific immune responses play essential roles in the process of MTB elimination by a host. Therefore, the activation of NK cells, macrophages, B cells, CD4^+^ T cells (Th1 and Th2 cells), and CD8^+^ T cells (CTLs) induced by the PP19128R vaccine was predicted by the C-IMMSIM server. Our results showed that the PP19128R vaccine activated NK cells and maintained their number at between 310 and 380 cells/mm^3^ (Figure 5A). Interestingly, the PP19128R vaccine activated macrophages and dendritic cells and led to their proliferation and differentiation. Immune simulation induced three proliferation peaks in type 2-presenting macrophages (Figure 5B) and dendritic cells (Figure 5C). However, unlike dendritic cells, the number of quiescent and active macrophages remained stable at 90 cells/mm^3^ for 90 days after the first immune simulation with the PP19128R vaccine but increased sharply after day 90, while the number of active macrophages decreased sharply and finally remained at 15 cells/mm^3^ (Figure 5B). In addition, the PP19128R vaccine induced significantly high levels of epithelial cells (Figure 5D). Similar to macrophages and dendritic cells, the PP19128R vaccine stimulated the differentiation and proliferation of B lymphocytes, resulting in three peaks in type 1-presenting and active B lymphocyte numbers at day 40 (600 cells/mm^3^), day 70 (710 cells/mm^3^), and day 110 (470 cells/mm^3^), respectively, after the first immune simulation (Figure 5E). Similar trends were observed for immunoglobulins and immune complexes (Figure 5F).

In addition, we analyzed the simulated adaptive immune responses induced by the PP19128R vaccine. The results showed that immune simulation of the PP19128R vaccine induced three peaks in the number of memory HTLs; the number increased up to 12,000 cells/mm^3^ after the third immunization especially (Figure 6A). In addition, we also found that PP19128R injection stimulated three peaks of active HTLs on days 20, 40, and 75 after the first immunization (Figure 6B). However, unlike for HTLs, PP19128R was less able to stimulate the immune system to generate memory CTLs (Figure 6C), with active CTLs peaking at day 50 after the first immunization (900 cells/mm^3^) and then gradually decreasing, while resting cytotoxic T lymphocytes showed the exact opposite trend (Figure 6D). Interestingly, we found that the PP19128R vaccine induced the differentiation of T lymphocytes into Th1-type lymphocytes and mediated a robust Th1-type immune response (Figure 6E). In addition, we observed that the PP19128R vaccine could induce regulatory T cells to peak rapidly (155 cells/mm^3^) after the first immunization and then gradually decline (Figure 6F). Finally, we also analyzed the ability of the PP19128R vaccine to induce IFN-γ production by immune cells. We found that three immunizations with PP19128R could induce the cellular cytokines IFN-γ (41,000 ng/mL, 400,000 ng/mL, and 380,000 ng/mL) and IL-2 (200,000 ng/mL, 690,000 ng/mL, and 480,000 ng/mL) to form three peaks (Figure 7).

### 3.5. The PP19128R Vaccine Induced Significantly Higher Levels of Immune Responses In Vitro

The recombinant plasmid of the PP19128R vaccine is shown in Figure 8A, and the recombinant protein with a molecular weight of 98.57 kDa was successfully expressed in *E. coli* and purified by SDS-PAGE (Figure 8B).

To verify the relevance and consistency of the immune profile of the PP19128R vaccine in silico and in vitro, we stimulated peripheral blood PBMCs from HCs and LTBI and ATB patients with the PP19128R vaccine in vitro. Our previously developed TB vaccine HP13138PB was used as a positive control. In addition, we analyzed its immunogenicity using ELISpot and CBA assays. The results showed that the number of IFN-γ^+^ T lymphocytes induced by the PP19128R vaccine was higher than that of IFN-γ^+^ T lymphocytes induced by AIM or the HP13138PB vaccine in HCs (Figure 9A), ATB patients (Figure 9B), and individuals with LTBI (Figure 9C). Furthermore, although the number of IFN-γ^+^ T lymphocytes induced by the PP19128R vaccine was not statistically different from the numbers induced by the negative control (AIM) and the positive control (the HP13138PB vaccine), we observed that the number of IFN-γ^+^ T lymphocytes in the PP19128R group was higher than those in the negative and positive control groups in the HC, ATB, and LTBI groups. These results indicate that the PP19128R vaccine has broad immunogenicity in the human population.

In addition, we also evaluated the ability of the PP19128R vaccine to induce the production of cytokines, such as IL-2, TNF-α, IFN-γ, IL-4, IL-6, IL-10, and IL-17A, in three groups in vitro. The results showed that: (1) the levels of TNF-α induced by the PP19128R vaccine in HCs (*p* < 0.0001) and ATB patients (*p* = 0.0039) were significantly higher than those induced by the AIM medium in HCs (Figure 10A); (2) the level of IFN-γ induced by the PP19128R vaccine in HCs (*p* = 0.0019) was significantly higher than that induced by the AIM medium in HCs (Figure 10B); (3) the levels of IL-6 (Figure 10C) induced by the PP19128R vaccine in HCs (*p* = 0.0003), ATB patients (*p* = 0.0009), and individuals with LTBI (*p* = 0.0386) were significantly higher than that induced by the AIM medium in HCs; (4) the levels of IL-10 (Figure 10D) induced by the PP19128R vaccine in HCs (*p* = 0.0063), ATB patients (*p* = 0.0181), and individuals with LTBI (*p* = 0.0010) were significantly higher than that induced by the AIM medium in HCs. Furthermore, the actual detected concentrations of IL-2, IL-4, and IL-17A (data not shown) cytokines were lower than the theoretical values that could be detected by the CBA kit, so we did not analyze these three cytokine concentrations. Interestingly, we found that, compared with the AIM negative controls: (1) the PP19128R vaccine was able to induce significantly high levels of Th1-type cytokines, such as IFN-γ and TNF-α, and Th2-type cytokines, such as IL-6 and IL-10, in HCs; (2) the PP19128R vaccine was able to induce significantly high levels of the Th1-type cytokine TNF-α and Th2-type cytokines, such as IL-6 and IL-10, in ATB patients; (3) the PP19128R vaccine induced significantly high levels of Th2-type cytokines, such as IL-6 and IL-10, in individuals with LTBI. These data suggest that the PP19128R vaccine not only has great potential as a preventive vaccine in healthy populations but may also have some preventive and therapeutic effects in individuals with latent TB infection and TB patients.

### 3.6. Correlation Analysis and Simple Linear Regression Analysis of Cytokines Induced by the PP19128R Vaccine

To further explore the characteristics of cellular immune responses induced by the PP19128R vaccine, we also investigated the cytokines that played significant roles and the correlations among them based on principal component analysis and correlation analysis. The results showed the following: (1) In the healthy population, PP19128R-induced cytokines were clustered into three groups—IFN-γ, IL-6, and TNF-α as one group; IL-10, IL-4, and IL-17A as another group; and IL-2 as a separate group—but no significant correlation was observed between cytokines (Figure 11A). (2) In the ATB patients, PP19128R-induced cytokines were clustered into three families, similarly to the HCs, and TNF-α was positively correlated with IFN-γ (*p* = 0.024) and IL-6 (*p* = 0.007), IFN-γ was positively correlated with IL-6 (*p* = 0.016), and IL-10 was positively correlated with IL-17A (*p* = 0.035) (Figure 11B). (3) In the LTBI population, PP19128R-induced cytokines were clustered into two groups—IFN-γ, IL-2, IL-4, IL-17A, and TNF-α as one group and IL-6 and IL-10 as another group—and positive correlations were also found between IL-2 and TNF-α (*p* < 0.0001), IL-4 (*p* = 0.003), or IL-17A (*p* = 0.015); between TNF-α and IL-4 (*p* = 0.009) or IL-17A (*p* = 0.019); and between IL-10 and IL-6 (*p* = 0.018) (Figure 11C).

The cytokines for which the above correlations existed were further analyzed using simple linear regression, which revealed that: (1) in LTBI subjects, significant positive correlations could be observed between PP19128R-induced TNF-α and IL-2 (R^2^ = 0.965, *p* < 0.0001, Y = 0.006311*X − 0.1866, Figure 12A), IL-4 and IL-2 (R^2^ = 0.7855, *p* = 0.0036, Y = 0.7980*X − 1.781, Figure 12B), IL-4 and TNF-α (R^2^ = 0.7065, *p* = 0.0090, Y = 117.8*X − 224.6, Figure 12C), IL-6 and IL-10 (R^2^ = 0.6343, *p* = 0.0180, Y = 0.01476*X − 172.9, Figure 12D), IL-17A and IL-2 (R^2^ = 0.6574, *p* = 0.0146, Y = 0.2586*X − 0.5414, Figure 12E), and IL-17A and TNF-α (R^2^ = 0.6302, *p* = 0.0186, Y = 39.42*X − 48.09, Figure 12F) cytokines; (2) similarly, significant positive correlations were observed between PP19128R-induced IFN-γ and TNF-α (R^2^ = 0.6708, *p* = 0.0242, Y = 107.7*X − 523.7, Figure 13A), IL-6 and TNF-α (R^2^ = 0.7989, *p* = 0.0067, Y = 0.1831*X − 2862, Figure 13B), IL-6 and IFN-γ (R^2^ = 0.6224, *p* = 0.0350, Y = 7.554*X − 7.930, Figure 13C), and IL-17A and IL-10 (R^2^ = 0.7218, *p* = 0.0155, Y = 0.001323*X − 14.71, Figure 13D) cytokines.

## 4. Discussion

In our previous study, we successfully developed a multiepitope vaccine named MP3RT that consisted of six immunodominant HTL epitopes [27,31,50]. Our results indicated that the MP3RT vaccine could induce significantly higher levels of Th1-type cytokines, CD3^+^ IFN-γ^+^ T lymphocytes, and MP3RT-specific IgG antibodies [27]. However, we found that the protective efficacy of the MP3RT vaccine was not better than that of the BCG vaccine. One possible reason is that the MP3RT vaccine consists of only HTL epitopes without CTL and B-cell epitopes. TLR agonists and helper peptides were not incorporated into the vaccine design to improve the immunogenicity and targeting of the vaccine. CD8^+^ T cells and B cells have been reported to be important in eliminating and killing MTB [4,8,23,51]. In another previous study, we developed an MEV named HP13138PB against MTB infection in silico and verified its immunological properties in in vitro experiments [49]. However, the HP13138PB vaccine was designed with the addition of TLR2 agonist PSMα4 but not TLR4 agonist, so its immunogenicity was not further enhanced.

The PP19128R vaccine developed in this study fully covers HTL, CTL and B-cell epitopes to compensate for these deficiencies. The PP19128R vaccine contains 19 HTL epitopes, 12 CTL epitopes, and 8 B-cell epitopes and has the highest antigenicity and immunogenicity, as well as being non-allergenic and non-toxic. In addition, it has been reported that subunit vaccines have the disadvantages of easy degradation, weak immunogenicity, and an inability to target delivery [4,52]. Therefore, in this study, the TLR2 agonist PorB, the TLR4 agonist RS-09, and the helper peptide PADRE were added to the amino acid sequence of the PP19128R vaccine to overcome these limitations of subunit vaccines. TLRs are a group of receptors widely distributed in macrophages and dendritic cells that mediate recognition of and response to MTB antigens through pathogen-associated molecular patterns (PAMPs) [53,54,55,56]. Previous studies have shown that TLR2 can induce the production of proinflammatory cytokines to limit MTB replication while counteracting antimicrobial effector mechanisms through immune evasion [57]. In addition, TLR4 plays an active role in the induction of anti-tuberculosis immune response and participates in the killing and clearance of MTB by inducing macrophage apoptosis and downregulating the intensity of the immune response in TB patients [58]. The use of TLR2 agonist PorB and TLR4 agonist RS-09 in the PP19128R vaccine enables the vaccine to target antigen-presenting cells, such as macrophages and dendritic cells, improving vaccine delivery efficiency. In addition to adjuvants and agonists, linkers were carefully selected to enhance vaccine expression, correct folding, and stability [59]. For example, the AAY linker can affect the structural strength through the protease cleavage site, while the KK linker can maintain the immunogenicity of the epitope [60,61]. Therefore, the flexible linker GPGPG, the rigid linker AAY, and the KK linker were used to construct the PP19128R vaccine.

The physical and chemical properties and spatial structure of MEV are critical to its ability to undertake its biological and immunological functions [62]. Proteins with molecular mass less than 100,000 Da require an instability index <40 to achieve good structural stability [63]. Our results showed that the PP19128R vaccine has a molecular weight of 98557.86 Da, an instability index of 36.04 (<40), and a lipid index of 74.35, which means that the PP19128R vaccine has good structural and thermal stability and is not easily degraded. Furthermore, solubility is a physical property that must be considered in the expression and purification of therapeutic proteins because protein molecules must be soluble in aqueous protein matrices for diffusion and in vivo biological effects [64]. The predicted results showed that the solubility of the PP19128R vaccine was 0.900675, indicating that the vaccine has good solubility.

In the antigen-induced immune response, recognition and tight binding between antigens and receptors are crucial [65]. Molecular docking results indicated that the PP19128R vaccine could tightly bind to TLR2 (−1324.77 kcal/mol) and TLR4 receptors (−1278 kcal/mol), suggesting that the PP19128R vaccine had a good TLR2/4 receptor binding ability [66,67]. Relevant immunological simulation results supported these data. In the in silico analysis, the PP19128R vaccine was observed to stimulate the proliferation of innate immune cells (such as NK cells, DCs, and macrophages) and adaptive immune cells (such as TH, TC, RT, and B lymphocytes). Innate immune cells are the host’s first line of defense against MTB infection, and the macrophages, NK cells, and DCs will phagocytose and kill MTB [4,25]. Pulmonary DCs are the first line of defense against MTB invasion, recognizing MTB through DC-specific intercellular adhesion molecule-3 predatory non-integrin (DC-SIGN) and TLR, which is followed by massive secretion of IL-1α, IL-1β, IL-10, and inducible nitric oxide synthase (iNOS) to kill MTB [68,69,70]. In the in silico analysis, three increasing peaks for the type 2-presenting DC and macrophage population were observed after three injections of the PP19128R vaccine, followed by three peaks in the IL-10 concentration. These data were consistent with the in vitro CBA assay results showing that stimulation with PP19128R induced significantly higher levels of IL-10 in HCs, ATB patients, and individuals with LTBI.

DCs migrate to peripheral lymph nodes and present MTB antigens to T lymphocytes [71,72]. As a result, T lymphocytes are activated and differentiated into IFN-γ^+^ Th1 lymphocytes (CD4^+^ T cells), cytotoxic T lymphocytes (CD8^+^ T cells), Th17 cells, Th2 cells, and regulatory T cells (Tregs) [73,74]. These activated effector cells enter the circulation, migrate to the site of MTB infection, and participate in local anti-tuberculosis immunity [75]. This study found that the PP19128R vaccine could significantly stimulate the proliferation of memory helper T, active T, cytotoxic T, and B lymphocytes and induce Th0 lymphocytes to differentiate into Th1 T lymphocytes, which secreted high levels of Th1-type cytokines or antibodies in silico. In addition, the CBA assay results showed that the PP19128R vaccine stimulated PBMCs derived from HCs, LTBI individuals, and ATB patients to produce significantly high levels of cytokines, such as IL-4, IL-6, and IL-10. These results suggest that the PP19128R vaccine may be a promising MTB vaccine capable of inducing potent cellular and humoral immune responses in healthy people, individuals with LTBI, and ATB patients. At the same time, these data also preliminarily suggest that the PP19128R vaccine not only has a potential preventive effect in healthy populations but may also have some preventive and therapeutic effects in LTBI individuals and patients with active TB. In addition, we found a correlation between PP19128R vaccine-induced cytokines in ATB patients and individuals with LTBI. The results of the linear regression analysis showed that the correlation between IL-2 and TNF-α in the LTBI population was as good as R^2^ = 0.9655, suggesting a significant positive correlation between IL-2 and TNF-α in the process of immune response to latent infection.

There are several limitations to our research: (1) the physicochemical properties and spatial structure of the PP19128R vaccine were only predicted in silico and not analyzed in the real world; (2) the subtypes of innate and adaptive immune cells were investigated in silico but not by flow cytometry; (3) the immunogenicity of the PP19128R vaccine was only validated in vitro with PBMCs from HCs, ATB patients, and LTBI participants and not in in vivo experiments in animal models; (4) the protective efficacy of the PP19128R vaccine was not evaluated in animal models.

## 5. Conclusions

In conclusion, this study developed a promising MEV against LTBI named PP19128R consisting of 19 HTL epitopes, 12 CTL epitopes, 8 B-cell epitopes, TLR agonists, and helper peptides. In the in silico analysis, the PP19128R vaccine showed excellent antigenicity, immunogenicity, non-toxicity, and no sensitization, and it can potentially target TLR2 and TLR4 on macrophages and activate T and B lymphocytes to produce high levels of cytokines, especially IFN-γ, TGF-β, and IL-2. The results of the ELISpot and CBA assays showed that the PP19128R vaccine could induce a higher level of IFN-γ^+^ T lymphocytes and stimulate the secretion of essential cytokines, such as IFN-γ, TNF-α, IL-6, and IL-10, in PBMCs obtained from HCs, ATB patients, and LTBI participants. This study provides a vaccine candidate for the prevention of LTBI in the future.

## Figures and Tables

**Figure 1 vaccines-11-00856-f001:**
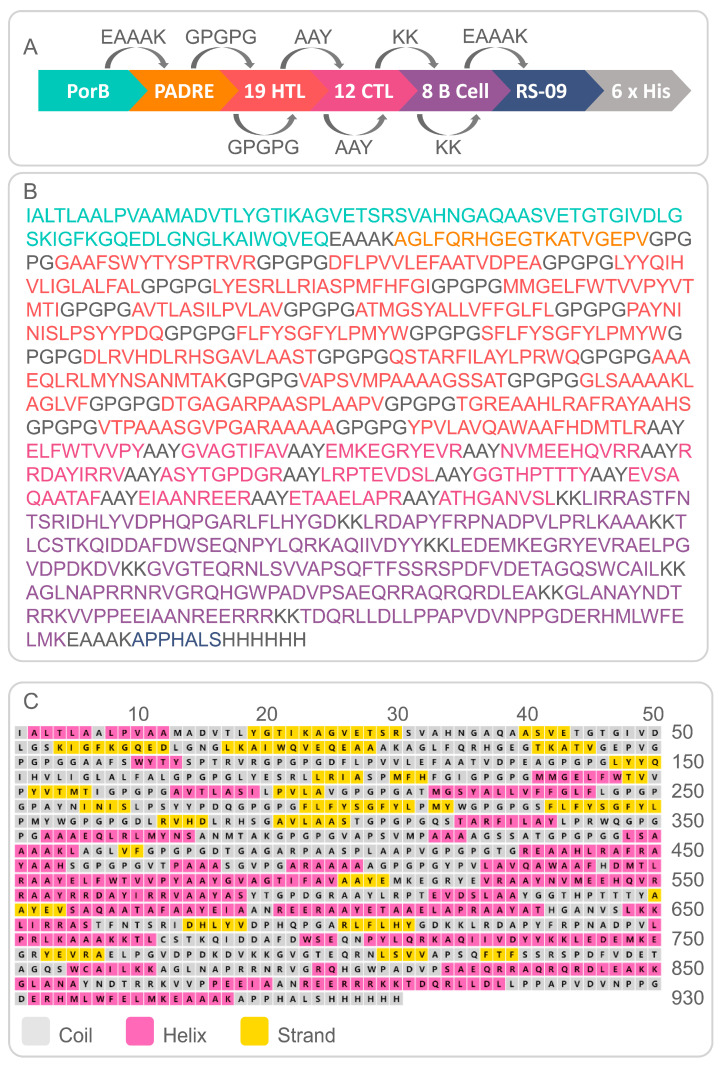
Schematic diagram of the PP19128R vaccine construction strategy and secondary structure. (**A**) The PP19128R vaccine was constructed based on TLR2 agonist PorB, pan HLA DR-binding epitope PADRE, 19 HTL epitopes, 12 CTL epitopes, 8 B-cell epitopes, and TLR-4 agonist RS-09. (**B**) The amino acid sequence of the PP19128R vaccine, with different elements indicated by different colors. (**C**) The coil, helix, and strand secondary structures in the amino acid sequence of the PP19128R vaccine are shown in gray, pink, and yellow, respectively.

**Figure 2 vaccines-11-00856-f002:**
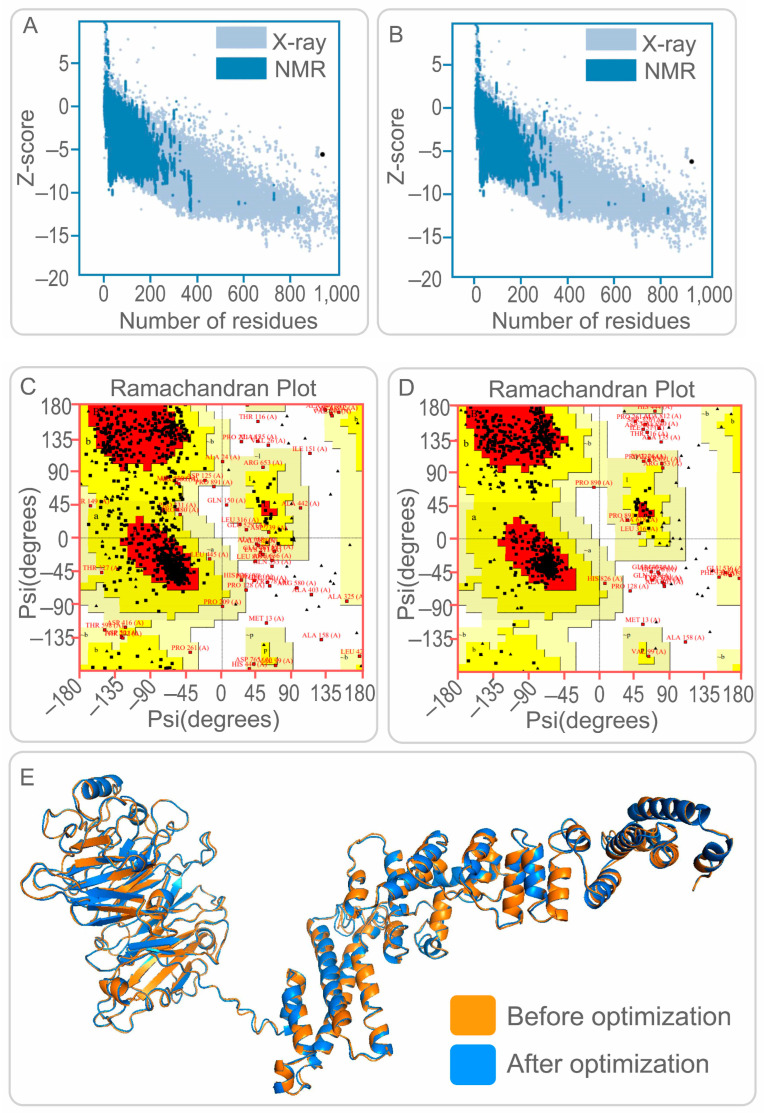
Construction and optimization of 3D models for PP19128R vaccine. (**A**,**B**) Z-score of the PP19128R vaccine predicted by ProSA web server. The Z-score of the 3D model of the PP19128R vaccine was −5.59 before optimization (**A**) and −6.28 after optimization (**B**). (**C**,**D**) Ramachandran plot of the PP19128R vaccine predicted by UCLA-DOE LAB-SAVES v6.0 server. The core, allow, gener, and disall of the PP19128R vaccine were 70.8%, 22.8%, 4.4%, and 2.0% before optimization (**C**) and 87.2%, 9.0%, 1.6%, and 2.2% after optimization (**D**). (**E**) The 3D structure of the PP19128R vaccine was constructed by the I-TASSER server and optimized by the Galaxy Refine server. Before optimization: orange; after optimization: blue.

**Figure 3 vaccines-11-00856-f003:**
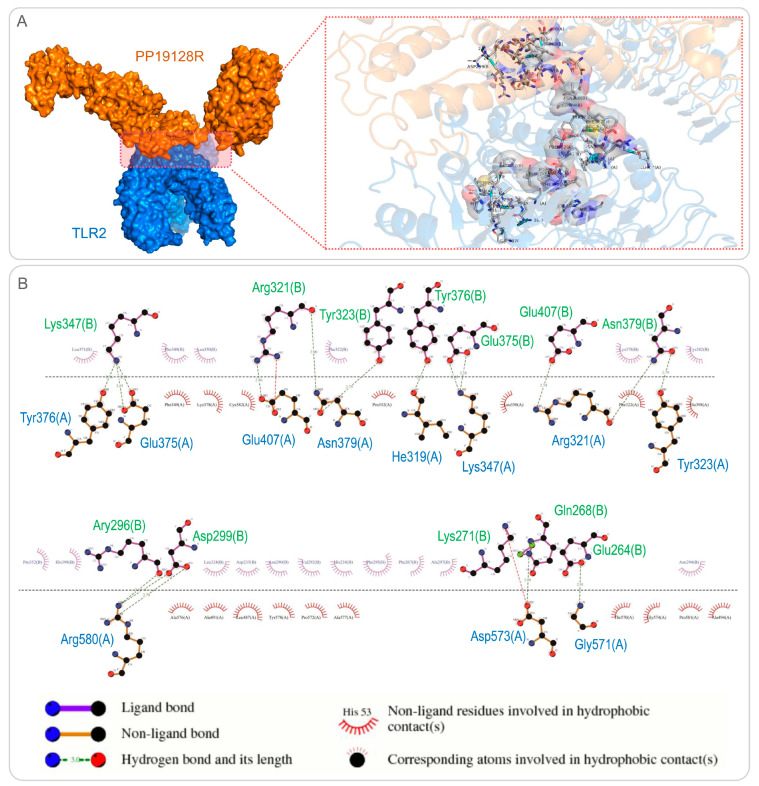
Molecular docking and binding site analysis for the PP19128R vaccine and TLR2. (**A**) Molecular docking between the PP19128R vaccine (in yellow) and TLR2 (in blue) was predicted by using the Cluspro server. A cartoon illustration of molecular docking is shown in the left panel. A 3D illustration of the interactions between bonds at the molecular docking site is shown in the right panel. (**B**) A 2D illustration of the binding sites between the PP19128R vaccine and TLR2 predicted by LigPlot+ v.2.2 (https://www.ebi.ac.uk/thornton-srv/software/LigPlus/ (accessed on 18 February 2022).

**Figure 4 vaccines-11-00856-f004:**
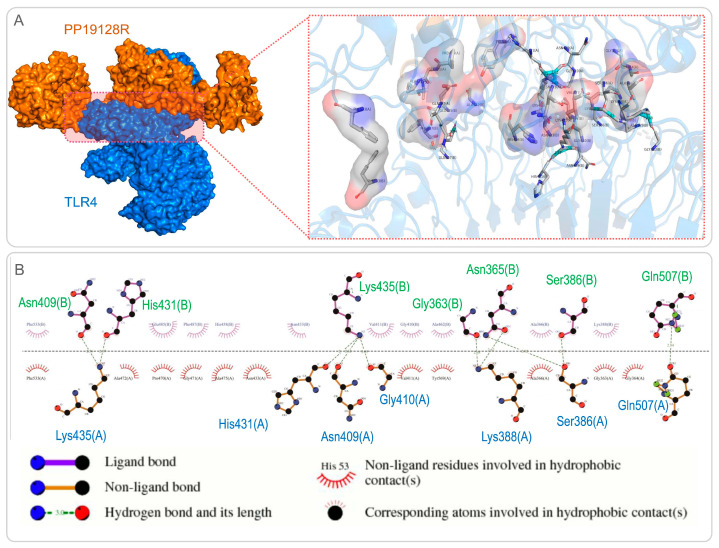
Molecular docking and binding site analysis for the PP19128R vaccine and TLR4. (**A**) Molecular docking between the PP19128R vaccine (in yellow) and TLR4 (in blue) was predicted by using the Cluspro server. A cartoon illustration of molecular docking is shown in the left panel. A 3D illustration of the interactions between bonds at the molecular docking site is shown in the right panel. (**B**) A 2D illustration of the binding sites between the PP19128R vaccine and TLR4 predicted by LigPlot+ v.2.2 (https://www.ebi.ac.uk/thornton-srv/software/LigPlus/ (accessed on 18 February 2022).

**Figure 5 vaccines-11-00856-f005:**
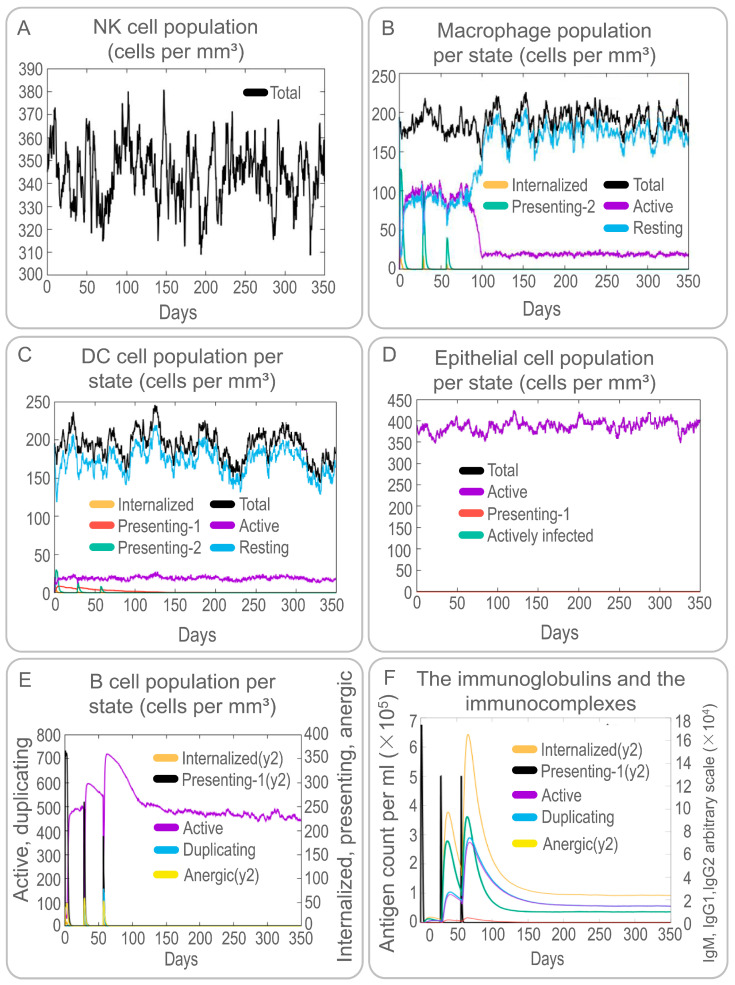
The PP19128R vaccine-induced changes in the populations of innate immune cells in silico. Immune simulation with the PP19128R vaccine was performed three times with the C-immune server in silico, and the populations of NK cells (**A**), macrophages (**B**), DCs (**C**), epithelial cells (**D**), B cells (**E**), and immunoglobulins and immune complexes (**F**) were analyzed.

**Figure 6 vaccines-11-00856-f006:**
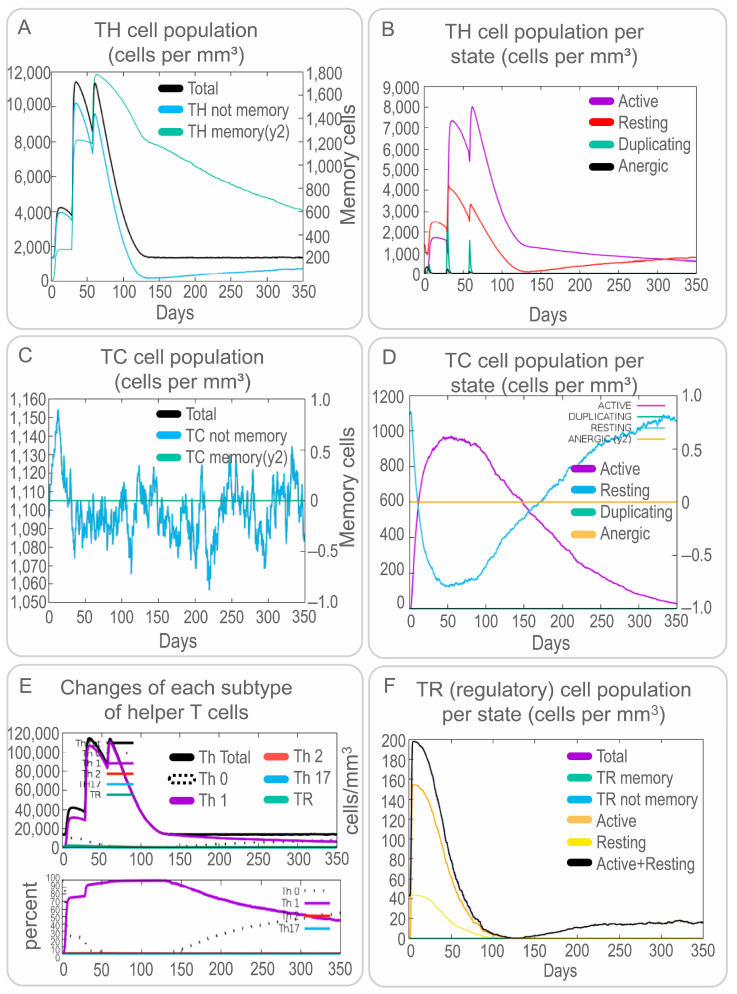
The PP19128R vaccine-induced changes in the populations of adaptive immune cells in silico. Simulated immunization with the PP19128R vaccine was performed three times by the C-immune server in silico, and the populations of helper T cells (**A**,**B**), cytotoxic T cells (**C**,**D**), subtypes of helper T cells (**E**), and regulatory T cells (**F**) were analyzed.

**Figure 7 vaccines-11-00856-f007:**
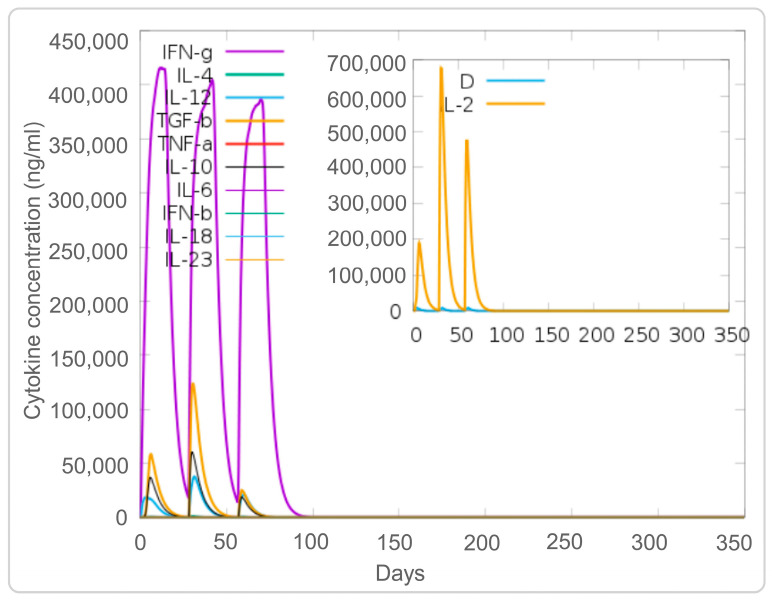
Cytokines induced by PP19128R vaccination. Simulated immunization with the PP19128R vaccine was performed three times by the C-immune server in silico, and the levels of cytokines were analyzed.

**Figure 8 vaccines-11-00856-f008:**
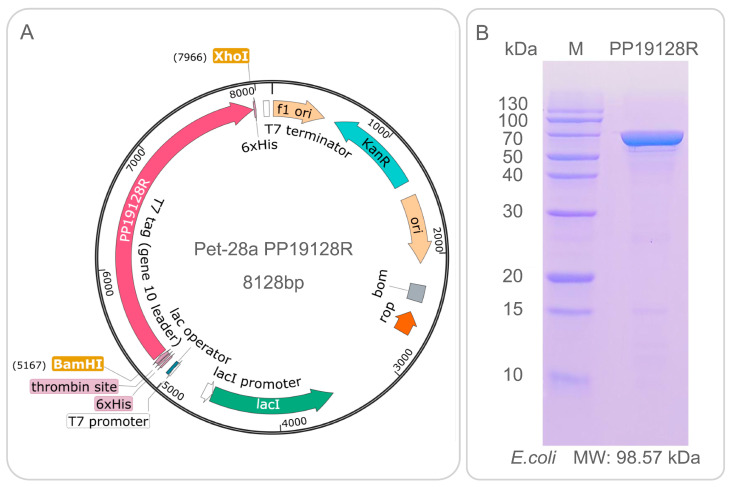
Construction of PP19128R-pET-28(a) recombinant plasmid and expression and purification of the PP19128R vaccine. The PP19128R-pET28(a) recombinant plasmid was constructed by inserting the nucleotide sequence of the PP19128R vaccine from the XhoI and BamHI restriction sites into the pET28(a) plasmid using SnapGene version 6.2.1 software (**A**). The PP19128R vaccine was then expressed in *E. coli*, purified through the C-terminal 6-his tag, and analyzed by SDS-PAGE (**B**).

**Figure 9 vaccines-11-00856-f009:**
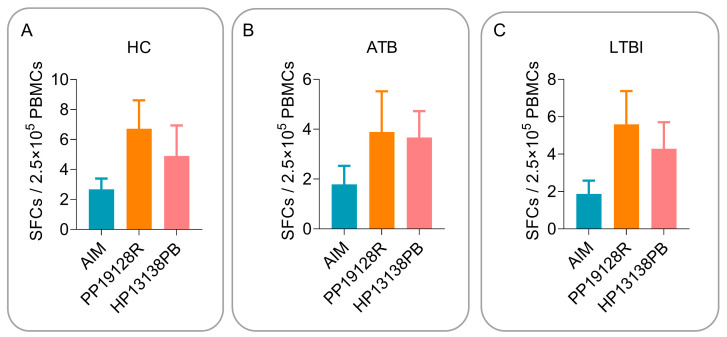
The number of PP19128R-specific IFN-γ^+^ T lymphocytes. The purified PP19128R vaccine (experiment group), AIM (negative control), and our previously designed vaccine HP13138PB [50] (positive control) were used to stimulate PBMCs collected from HCs (**A**), ATB patients (**B**), and LTBI individuals (**C**) in vitro. The number of antigen-specific IFN-γ^+^ T lymphocytes was detected by ELISpot. SFCs: the number of spot-forming cells; HCs: healthy controls; ATB: active tuberculosis; LTBI: latent tuberculosis infection.

**Figure 10 vaccines-11-00856-f010:**
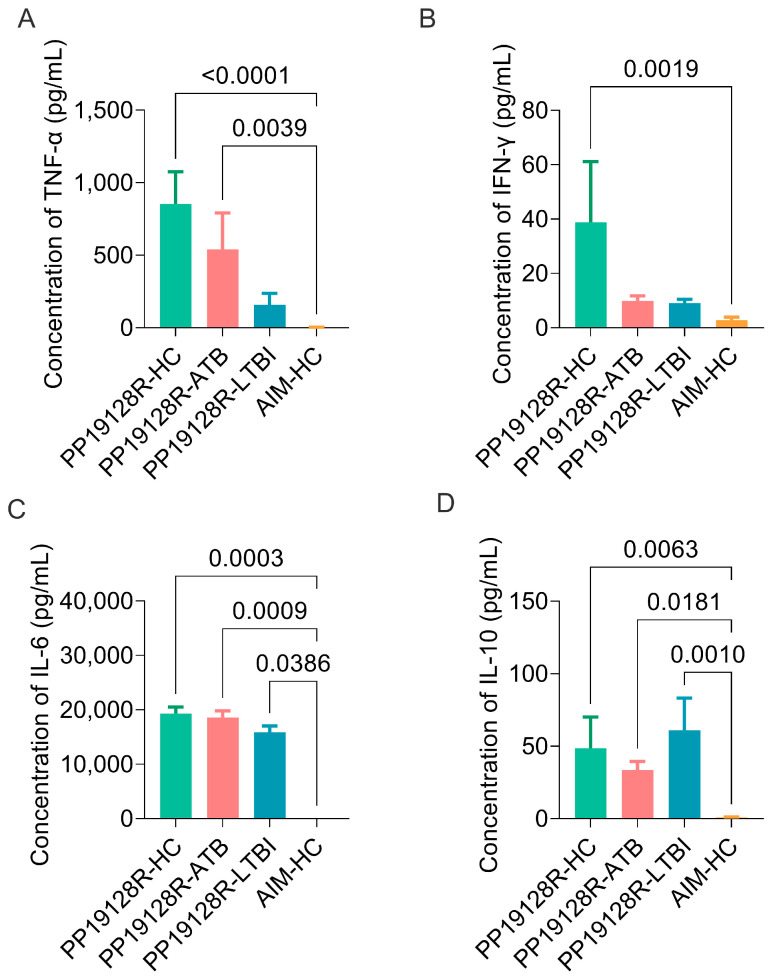
The levels of cytokines induced by the PP19128R vaccine. The purified PP19128R vaccine was used to stimulate PBMCs collected from HCs, ATB patients, and LTBI individuals in vitro. The levels of TNF-α (**A**), IFN-γ (**B**), IL-6 (**C**), and IL-10 (**D**) cytokines were detected with a BD CBA Human Th1/Th2/Th17 Cytokine Kit.

**Figure 11 vaccines-11-00856-f011:**
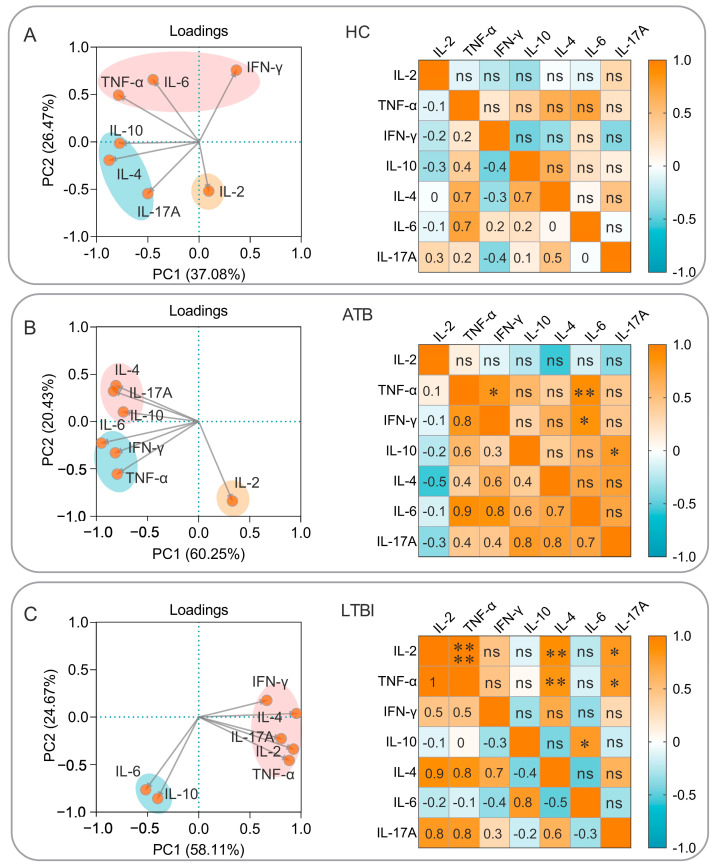
Principal component analysis (PCA) and correlation analysis. The purified PP19128R vaccine was used to stimulate PBMCs obtained from HCs (**A**), ATB patients (**B**), and individuals with LTBI (**C**). PCA and correlation analysis were performed to analyze the potential effects among cytokines using GraphPad Prism 9.5.1 software. The PCA analysis selected PCs with eigenvalues greater than 1.0 (“Kaiser rule”) as the component selection method. The proportions of variance for PC1 and PC2 are plotted on the x-axis and y-axis, respectively. The correlation analysis used Pearson’s R to determine the relationship between two cytokines, and R values of −1 to 1 are shown in blue to orange, respectively. *, *p* < 0.05, **, *p* < 0.01, ****, *p* < 0.0001; ns, no significant difference.

**Figure 12 vaccines-11-00856-f012:**
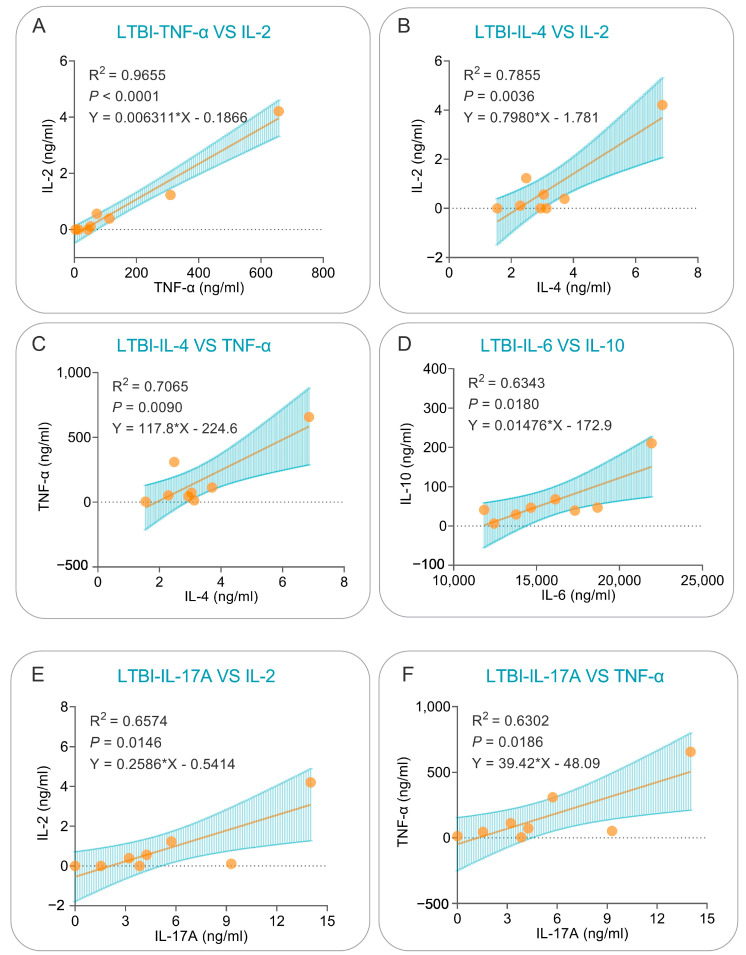
Simple linear regression analysis of cytokines in the LTBI population. Correlations for TNF-α vs. IL-2 (**A**), IL-4 vs. IL-2 (**B**), IL-4 vs. TNF-α (**C**), IL-6 vs. IL-10 (**D**), IL-17A vs. IL-2 (**E**), and IL-17A vs. TNF-α (**F**) were analyzed using simple linear regression in GraphPad Prism 9.5.1 software. The R-squared values for the goodness of fit, *p* values, and equations are shown in each panel. A *p* value lower than 0.05 was considered significant.

**Figure 13 vaccines-11-00856-f013:**
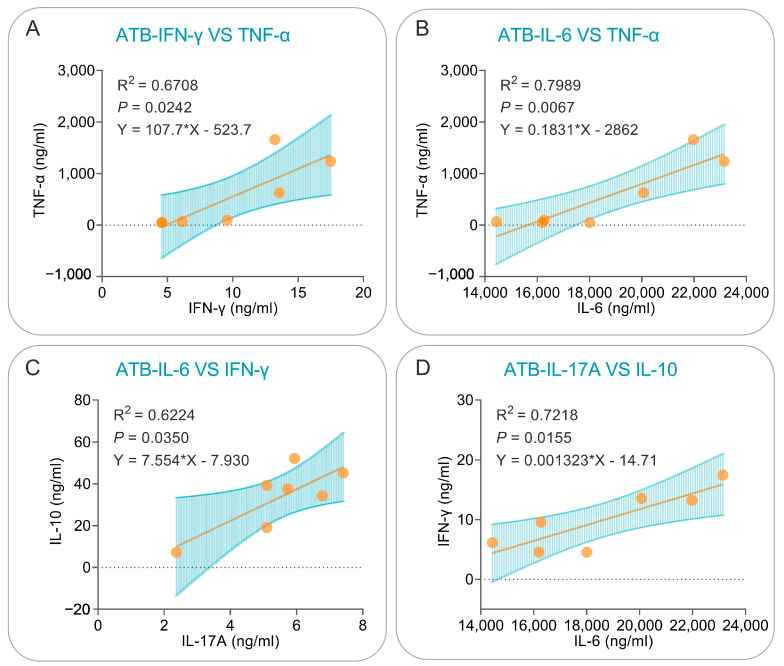
Simple linear regression analysis of cytokines in the ATB population. Correlations for IFN-γ vs. TNF-α (**A**), IL-6 vs. TNF-α (**B**), IL-6 vs. IFN-γ (**C**), and IL-17A vs. IL-10 (**D**) were analyzed using simple linear regression in GraphPad Prism 9.5.1 software. The R-squared values for the goodness of fit, *p* values, and equations are shown in each panel. A *p* value lower than 0.05 was considered significant.

**Table 1 vaccines-11-00856-t001:** The immunodominant HTL, CTL, and B-cell epitopes of the PP19128R vaccine.

Antigen	Allele	Peptide	Adjusted Rank	Antigenicity Score	IFN-γ Score	Immunogenicity Score	ABC Score	Allergenicity *		Toxin Pred
**HTL**										
Rv1736c	HLA-DQA1*02:01, DQB1*05:02	GAAFSWYTYSPTRVR	0.14	0.9205	0.43799388	NA	NA	NO	NO	NT
	HLA-DQA1*05:01, DQB1*02:01	DFLPVVLEFAATVDPEA	0.31	1.0044	0.60486444	NA	NA	NO	NO	NT
	HLA-DRB1*15:02	LYYQIHVLIGLALFAL	0.3	0.9211	0.43948812	NA	NA	NO	NO	NT
	HLA-DRB1*14:04	LYESRLLRIASPMFHFGI	0.38	0.7205	0.33801445	NA	NA	NO	NO	NT
	HLA-DRB1*15:02	MMGELFWTVVPYVTMTI	0.39	0.7048	0.34250203	NA	NA	NO	NO	NT
Rv1737c	HLA-DQA1*06:01, DQB1*03:03	AVTLASILPVLAV	0.34	0.8199	0.38907504	NA	NA	NO	NO	NT
	HLA-DRB1*15:02	ATMGSYALLVFFGLFL	0.46	0.856	0.72004245	NA	NA	NO	NO	NT
Rv1980c	HLA-DRB3*02:02	PAYNINISLPSYYPDQ	0.08	1.3614	2	NA	NA	NO	NO	NT
Rv1981c	HLA-DRB1*15:02	FLFYSGFYLPMYW	0.08	1.3283	1	NA	NA	NO	NO	NT
	HLA-DQA1*02:01, DQB1*05:02		0.12	1.3283	1	NA	NA	NO	NO	NT
	HLA-DQA1*01:01, DQB1*05:01		0.16	1.3283	1	NA	NA	NO	NO	NT
	HLA-DPA1*01, DPB1*04:01		0.19	1.3283	1	NA	NA	NO	NO	NT
	HLA-DRB1*15:02	SFLFYSGFYLPMYW	0.06	1.072	1	NA	NA	NO	NO	NT
	HLA-DPA1*01, DPB1*04:01		0.14	1.072	1	NA	NA	NO	NO	NT
	HLA-DQA1*02:01, DQB1*05:02		0.16	1.072	1	NA	NA	NO	NO	NT
	HLA-DQA1*01:01, DQB1*05:01		0.19	1.072	1	NA	NA	NO	NO	NT
	HLA-DPA1*01:03, DPB1*02:01		0.31	1.072	1	NA	NA	NO	NO	NT
Rv2659c	HLA-DQA1*05:01, DQB1*03:01	DLRVHDLRHSGAVLAAST	0.44	0.9292	2	NA	NA	NO	NO	NT
Rv3429	HLA-DQA1*02:01, DQB1*05:02	QSTARFILAYLPRWQ	0.28	0.7639	0.13606656	NA	NA	NO	NO	NT
	HLA-DRB3*02:02	AAAEQLRLMYNSANMTAK	0.41	0.7112	0.54792199	NA	NA	NO	NO	NT
Rv3873	HLA-DQA1*03:01, DQB1*06:01	VAPSVMPAAAAGSSAT	0.31	1.0173	0.23735279	NA	NA	NO	NO	NT
	HLA-DQA1*03:01, DQB1*06:01	GLSAAAAKLAGLVF	0.43	0.7126	0.67278691	NA	NA	NO	NO	NT
	HLA-DQA1*05:01, DQB1*03:01	DTGAGARPAASPLAAPV	0.15	0.8414	0.82589324	NA	NA	NO	NO	NT
Rv3879	HLA-DQA1*02:01, DQB1*05:02	TGREAAHLRAFRAYAAHS	0.03	0.7317	1.4848492	NA	NA	NO	NO	NT
	HLA-DQA1*05:01, DQB1*03:01	VTPAAASGVPGARAAAAA	0.47	0.8376	1.1311669	NA	NA	NO	NO	NT
	HLA-DQA1*02:01, DQB1*05:02	YPVLAVQAWAAFHDMTLR	0.44	0.7948	0.84318227	NA	NA	NO	NO	NT
**CTL**										
Rv1736c	HLA-B*15:02, HLA-B*15:11	ELFWTVVPY	NA	1.3999	NA	0.36591	NA	NO	NO	NT
Rv1737c	HLA-A*02:01	GVAGTIFAV	NA	0.9924	NA	0.33154	NA	NO	NO	NT
Rv2031c	HLA-A*33:03	EMKEGRYEVR	NA	1.678	NA	0.21929	NA	NO	NO	NT
Rv2626c	HLA-A*33:03	NVMEEHQVRR	NA	0.8767	NA	0.12879	NA	NO	NO	NT
Rv2656c	HLA-C*06:02, HLA-C*07:02	RRDAYIRRV	NA	0.9325	NA	0.24217	NA	NO	NO	NT
Rv2659c	HLA-A*11:01	ASYTGPDGR	NA	1.0059	NA	0.09894	NA	NO	NO	NT
Rv1511	HLA-C*07:02, HLA-C*06:02	LRPTEVDSL	NA	1.2094	NA	0.09388	NA	NO	NO	NT
Rv1980c	HLA-B*15:01	GGTHPTTTY	NA	1.7475	NA	0.12633	NA	NO	NO	NT
Rv3872	HLA-B*15:11	EVSAQAATAF	NA	0.8759	NA	0.00342	NA	NO	NO	NT
Rv3425	HLA-A*33:03	EIAANREER	NA	1.0439	NA	0.23749	NA	NO	NO	NT
Rv3878	HLA-A*33:03	ETAAELAPR	NA	1.0877	NA	0.16567	NA	NO	NO	NT
Rv3879	HLA-C*03:04, HLA-C*01:02, HLA-A*30:01	ATHGANVSL	NA	1.8216	NA	0.01479	NA	NO	NO	NT
**B cell**										
Rv1511	NA	LIRRASTFNTSRIDHLYVDPHQPGARLFLHYGD	NA	NA	NA	NA	0.77	NO	NO	NT
Rv1737c	NA	LRDAPYFRPNADPVLPRLKAAA	NA	NA	NA	NA	0.78	NO	NO	NT
Rv1981c	NA	TLCSTKQIDDAFDWSEQNPYLQRKAQIIVDYY	NA	NA	NA	NA	0.81	NO	NO	NT
Rv2031c	NA	LEDEMKEGRYEVRAELPGVDPDKDV	NA	NA	NA	NA	0.94	NO	NO	NT
Rv2660c	NA	GVGTEQRNLSVVAPSQFTFSSRSPDFVDETAGQSWCAIL	NA	NA	NA	NA	0.86	NO	NO	NT
Rv2653c	NA	AGLNAPRRNRVGRQHGWPADVPSAEQRRAQRQRDLEA	NA	NA	NA	NA	0.92	NO	NO	NT
Rv3425	NA	GLANAYNDTRRKVVPPEEIAANREERRR	NA	NA	NA	NA	0.86	NO	NO	NT
Rv3879	NA	TDQRLLDLLPPAPVDVNPPGDERHMLWFELMK	NA	NA	NA	NA	0.82	NO	NO	NT

* AllerTOP v.2.0 and Allergen FP v.1.0 were used to predict allergenicity. NA: not applicable; NT: non-toxin; NO: non-allergenic.

**Table 2 vaccines-11-00856-t002:** Global population coverage of the PP19128R vaccine for HLA I and II alleles.

Population/Area	Class I			Class II		
	Coverage ^a^	Average_hit ^b^	Pc90 ^c^	Coverage ^a^	Average_hit ^b^	Pc90 ^c^
Central Africa	57.38%	1.32	0.23	82.75%	2.79	0.58
East Africa	61.10%	1.2	0.26	89.22%	3.63	0.93
East Asia	85.63%	2.29	0.7	73.85%	2.72	0.38
Europe	82.00%	1.76	0.56	99.06%	4.41	1.4
North Africa	65.52%	1.42	0.29	80.01%	4.04	0.5
North America	82.17%	1.88	0.56	99.93%	5.05	2.94
Northeast Asia	92.53%	2.66	1.12	97.97%	5.07	2.19
Oceania	76.33%	1.38	0.42	97.55%	4.78	2.05
South America	65.56%	1.29	0.29	98.12%	4.69	2.37
South Asia	70.59%	1.83	0.34	97.75%	5.68	1.65
Southeast Asia	91.15%	2.63	1.05	88.86%	3.84	0.9
Southwest Asia	64.29%	1.34	0.28	83.80%	4.01	0.62
West Africa	59.42%	1.33	0.25	88.35%	4.03	0.86
World	82.24%	1.88	0.56	93.71%	4.16	1.25

^a^ Projected population coverage. ^b^ Average number of epitope hits/HLA combinations recognized by the population. ^c^ Minimum number of epitope hits/HLA combinations recognized by 90% of the population.

## Data Availability

All data generated or analyzed during this study are included in the published article.

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
