# Peer review of "PP19128R, a Multiepitope Vaccine Designed to Prevent Latent Tuberculosis Infection, Induced Immune Responses In Silico and In Vitro Assays"

_vaccines, 2023, doi:10.3390/vaccines11040856_

Round 1

Reviewer 1 Report

I found the paper di Jiang et al  at times very interesting and in other points a little too complex. I believe that the studies made and the results obtained are encouraging and I believe that the authors have been honest in pointing out all the critical aspects of the work in the final part of the results. I have nothing to add except the absence of the standard deviation in figure 10 on the LTBI histogram.

Author Response

Responses to Reviewer 1:

1. I found the paper di Jiang et al at times very interesting and in other points a little too complex. I believe that the studies made and the results obtained are encouraging and I believe that the authors have been honest in pointing out all the critical aspects of the work in the final part of the results. I have nothing to add except the absence of the standard deviation in figure 10 on the LTBI histogram.

Response: Thank you very much for your valuable comments and suggestions. We attach great importance to your suggestions, which will be crucial to improving the level and quality of our paper. As you mentioned in your comments: “the absence of the standard deviation in figure 10 on the LTBI histogram”. We apologize for the fact that the error bars of the LTBI group data in Figure 10 in our original manuscript were not clearly displayed because the color was too light. According to your suggestion, we have redrawn Figure 10 and bolded the error bar to make it more intuitive.

Figure 10

Reviewer 2 Report

General comments

            The present paper is an extension of previous works by the same authors in relation to the development of a new-brand synthetic vaccine,  to prevent Mycobacterium tuberculosis disease, better than the well-known BCG vaccine.

            It is a paper of significance in the field,

            This work is based on a good number of diverse in silico methodologies and on a few in vitro experimental findings. The authors are aware of the limitations of their research such as the lack of in vivo observations.

            While in general terms their observations point to the usefullness of the vaccine for the intended purpose, there are some aspects which warrant a better description.

Specific comments

1.      There is a confusion between activation and proliferation of T cells. Actually, the “ ELISpot Pro kit is intended for the enumeration of cells secreting human IFN-γ using the ELISpot assay”, not for an estimation of their proliferation. In the Abstract/ Results, it is stated that “In vitro experiments showed that the PP19128R vaccine significantly increased the number of interferon-gamma positive (IFN-.+) T lymphocytes”, which is supported by the experimental observations made, but in Conclusions “The results of ELISPOT… assay showed that the PP19128R 588 vaccine could induce the proliferation of IFN-γ+ T lymphocytes…”  this statement is not supported by that applied assay. The authors should modify this reference to the proliferation of T lymphocytes.

2.      The cytokine profiles are presented and described in such a way that there is not a clear  distinction, if any, between TBI and LTBI patients. The authors should provide a more precise description of those profiles, comparing . side by side, TBI and LTBI patients,  and a better discussion of their meaning in the context of the intended usefullness of the vaccine under consideration Should this vaccine only usefull to prevent  the LTBI or as well the TBI disease?

Author Response

Responses to Reviewer 2

The present paper is an extension of previous works by the same authors in relation to the development of a new-brand synthetic vaccine, to prevent Mycobacterium tuberculosis disease, better than the well-known BCG vaccine. It is a paper of significance in the field. This work is based on a good number of diverse in silico methodologies and on a few in vitro experimental findings. The authors are aware of the limitations of their research such as the lack of in vivo observations. While in general terms their observations point to the usefullness of the vaccine for the intended purpose, there are some aspects which warrant a better description.

Response: Thank you very much for reviewing this article in the midst of your busy schedule. What you suggested will significantly improve the level of our manuscript, we have carefully read these comments and modified our draft according to your suggestions. Please find these modifications in the following sections. Please do not hesitate to reply to me in the submission system if you have any good suggestions for improving this manuscript. Changes have been shown in highlight in our revised manuscript.

1. There is a confusion between activation and proliferation of T cells. Actually, the “ ELISpot Pro kit is intended for the enumeration of cells secreting human IFN-γ using the ELISpot assay”, not for an estimation of their proliferation. In the Abstract/ Results, it is stated that “In vitro experiments showed that the PP19128R vaccine significantly increased the number of interferon-gamma positive (IFN-.+) T lymphocytes”, which is supported by the experimental observations made, but in Conclusions “The results of ELISPOT… assay showed that the PP19128R 588 vaccine could induce the proliferation of IFN-γ+ T lymphocytes…” this statement is not supported by that applied assay. The authors should modify this reference to the proliferation of T lymphocytes.

Response: Thank you. We are very sorry for this confusion in our original manuscript. We do agree with what you suggested that the “ELISpot Pro kit is intended for the enumeration of cells secreting human IFN-γ using the ELISpot assay”. We have corrected the sentences in the section of Conclusions “The results of ELISPOT… assay showed that the PP19128R 588 vaccine could induce the proliferation of IFN-γ+ T lymphocytes…”. Now, it was read as: “The results of ELISPOT and CBA assays showed that the PP19128R vaccine could in-duce significantly higher level of IFN-γ+ T lymphocytes and stimulate the secretion of important cytokines such as IFN-γ, TNF-α, IL-4, IL-6, and IL-10 in PBMCs obtained from HCs, ATB patients, and LTBI participants.” (Lines 589-592). Furthermore, we also updated all information related to what you have suggested above in our revised manuscript following your kind comments. These changes have been showing in the highlights.

2. The cytokine profiles are presented and described in such a way that there is not a clear distinction, if any, between TBI and LTBI patients. The authors should provide a more precise description of those profiles, comparing . side by side, TBI and LTBI patients, and a better discussion of their meaning in the context of the intended usefullness of the vaccine under consideration Should this vaccine only usefull to prevent the LTBI or as well the TBI disease?

Response: Thanks. We are very sorry for this carelessness in description of cytokine profiles. In this study, we enrolled there groups of population, including health controls, individuals with LTBI, and active TB patients (TBI you mentioned above). We detected the cytokine levels induced by the PP19128R vaccine and compared among three groups. We have added a more precise description of those profiles, comparing, and a better discussion of their potential usefulness in fighting against LTBI and TBI in our revised manuscript following your kind suggestion. Now, it was read as: “Interestingly, we found that compared with AIM-negative controls: (1) PP19128R vaccine was able to induce significantly high levels of Th1-type cytokines such as INF-γ and TNF-α and Th2-type cytokines such as IL-4, IL-6 and IL-10 in the healthy population; (2) PP19128R vaccine was able to induce significantly high levels of Th1-type cytokine TNF-α, Th2-type cytokines such as IL-4, IL-6 and IL-10, and the Th17-type cytokine IL-17A in primary active TB patients; (3) PP19128R vaccine induced significantly high levels of Th2-type cytokines such as IL-6 and IL-10, and the Th17-type cytokine IL-17A in individuals with LTBI. These data suggest that the PP19128R vaccine not only has great potential as a preventive vaccine in healthy population, but may also have some preventive and therapeutic effects in individuals with latent TB infection and TB patients.” (Lines 445-454).

Furthermore, we also gave a discussion in the section of Discussion for what you are mentioned above. You can find this discussion in lines 584-589, “These results suggest that PP19128R vaccine may be a promising MTB vaccine capable of inducing potent cellular and humoral immune responses in healthy people, individuals with LTBI, and ATB patients. At the same time, these data also preliminary suggest that the PP19128R vaccine not only has a potential preventive effect in the healthy population but also may have some preventive and therapeutic effect in LTBI individuals and patients with active TB.”

Reviewer 3 Report

Jiang et al. have attempted to answer a long-standing question in the field of tuberculosis vaccine research. Following the multiple epitope strategy, the authors have combined several epitopes from B- and T-cells and performed extensive in silico analysis to test the vaccine's antigenicity, immunogenicity, and solubility. The authors have also tested the ability of the vaccine to induce cytokine release by PBMCs isolated from the healthy controls, LTBI, and active TB patients. While the idea is mildly interesting, the study lacks the necessary controls and data to support the claims. 

Major comments

  1. The authors have molecular docking techniques to understand the binding of the vaccine with TLR2. The in silico analysis is speculative and needs further validation using the established biophysical assays to study binding affinities.

  2. Fig 9, the vaccine does show induction of IFNy for all three samples. However, there is no calculation of significance where the changes shown in the graph are statistically significant. There is no inclusion of positive control. Without a positive control, it is impossible to estimate the vaccine's effect. The authors should include a known Mtb antigen in the Elispot assay to reach any conclusion. 

  3. In Fig 10, the data presented is insufficient to support the claims made by the authors in the title. The vaccine is for sure inducing the production of the cytokines in the healthy controls primarily compared to the negative control. However, there is no positive control. The cell types used should have been T-cells considering the authors have access to the in vivo models. The immune response seems to be all over the place, which suggests that the cells are reacting to the presence of the foreign peptide. However, this does not support that the peptide is a vaccine candidate. The authors should use the standard preclinical method to test the potential vaccine's efficacy and safety. The assays are not sufficient to claim the effectiveness of the vaccine.

Author Response

Responses to Reviewer 3:

Jiang et al. have attempted to answer a long-standing question in the field of tuberculosis vaccine research. Following the multiple epitope strategy, the authors have combined several epitopes from B- and T-cells and performed extensive in silico analysis to test the vaccine's antigenicity, immunogenicity, and solubility. The authors have also tested the ability of the vaccine to induce cytokine release by PBMCs isolated from the healthy controls, LTBI, and active TB patients. While the idea is mildly interesting, the study lacks the necessary controls and data to support the claims.

Response: Thank you very much for your constructive comments and suggestions on our manuscript and for your affirmation of our work. We have carefully read your comments and suggestions and tried our best to modify and improve them to meet your requirements. Changes have been showed in highlight in our revised manuscript.

1. The authors have molecular docking techniques to understand the binding of the vaccine with TLR2. The in silico analysis is speculative and needs further validation using the established biophysical assays to study binding affinities.

Response: Thanks for your professional suggestion. In this study, we performed molecular docking to understand the binding of the PP19128R vaccine with TLR2 and TLR4. As the reviewer mentioned above, the in-silico analysis is speculative, but as an exploratory study based on bioinformatics and immunoinformatics techniques, in silico modeling of the dynamic relationship between vaccine and receptors is a routine and standardized approach, which can be validated in the following recently published references in the journal Vaccines (PMID: 36992295, PMID: 36851275, PMID: 36851219, and PMID: 36679917). As a vaccine, its success depends on the safety, immunogenicity, and protection efficiency of the vaccine. Therefore, we will first evaluate the safety, immunogenicity, and protection efficiency in animal models. Only if these criteria are met will we proceed to the study of spatial structure and biological effects. We agree with the reviewers and hope to use established biophysical assays to validate the binding affinity of the vaccine and the receptor in future work based on your suggestion, after confirming the protective effect of the vaccine in a mouse model. We believe that in this way we will not only reduce the cost of the assay but also save time.

2. Fig 9, the vaccine does show induction of IFNy for all three samples. However, there is no calculation of significance where the changes shown in the graph are statistically significant. There is no inclusion of positive control. Without a positive control, it is impossible to estimate the vaccine's effect. The authors should include a known Mtb antigen in the Elispot assay to reach any conclusion.

Response: Thank you very much for your kind comments. We have added a new group HP13138PB, a previously developed TB vaccine by our team, in Figure 9 in our revised manuscript following your constructive suggestion. The description of ELISPOT has been updated in the Results section. Now, it was read as: “To verify the relevance and consistency of the immune profile of the PP19128R vaccine in silico and in vitro, we stimulated peripheral blood PBMCs from HC, LTBI, and ATB patients with the PP19128R vaccine in vitro. Our previously developed TB vaccine HP13138PB was used as a positive control. In addition, we analyzed its immunogenicity by ELISPOT and CBA assays. The results showed that the number of IFN-γ+ T lymphocytes induced by the PP19128R vaccine was higher than that of IFN-γ+ T lymphocytes induced by AIM or HP13138PB vaccine in HCs (Figure 9A), ATB patients (Figure 9B), and individuals with LTBI (Figure 9C). Although the number of IFN-γ+ T lymphocytes induced by the PP19128R vaccine was not statistically different from that induced by the negative control AIM and positive control HP13138PB vaccine, we observed that the number of IFN-γ+ T lymphocytes in the PP19128R group was higher than that in the negative and positive control groups in the HC, ATB and LTBI groups. These results indicate that the PP19128R vaccine has broad immunogenicity in the human population.” (Lines 416-428)

Figure 9. The number of PP19128R-specific IFN-γ+ T lymphocytes. The purified PP19128R vaccine (experiment group), AIM (negative control), and our previously designed vaccine HP13138PB[50] (positive control) was used to stimulate PBMCs collected from HCs (A), ATB patients (B), and LTBI individuals (C) in vitro. The number of antigen-specific IFN-γ+ T lymphocytes were detected by ELISPOT. HCs: Healthy controls; ATB: active tuberculosis; LTBI: Latent tuberculosis infection.

3. In Fig 10, the data presented is insufficient to support the claims made by the authors in the title. The vaccine is for sure inducing the production of the cytokines in the healthy controls primarily compared to the negative control. However, there is no positive control. The cell types used should have been T-cells considering the authors have access to the in vivo models. The immune response seems to be all over the place, which suggests that the cells are reacting to the presence of the foreign peptide. However, this does not support that the peptide is a vaccine candidate. The authors should use the standard preclinical method to test the potential vaccine's efficacy and safety. The assays are not sufficient to claim the effectiveness of the vaccine.

Response: Thanks. We agree with your comments and suggestions stated above. Tuberculosis (TB) is an infectious disease that threatens the life and health of the world. However, there is only one vaccine used to prevent TB, namely Bacillus Calmette-Guerin (BCG) vaccine. BCG is a live attenuated vaccine containing several thousand antigens. In the evaluation of tuberculosis vaccines, BCG is often used as a positive control to compare the protective efficacy of the novel vaccine. The experiments in which BCG was used as a positive control were mainly those used to evaluate the protective efficiency of the vaccine, such as survival rate, colony count, pathological analysis, and so on. As an article focusing on the design of a tuberculosis vaccine based on bioinformatics and immunoinformatics technology, we have only conducted a preliminary evaluation of the immunogenicity of the vaccine, and have not used animal models to evaluate its immune protection efficiency. We will evaluate this vaccine in both humanized and wild-type animal models in future studies, with BCG as a positive control. In addition, according to your comments and suggestions, we have also redrawn Figure 10 to make it more accurate and intuitive to present our research results.

Figure 10. The levels of cytokines induced by the PP19128R vaccine. The purified PP19128R vaccine was used to stimulate PBMCs collected from HCs, ATB patients, and LTBI individuals in vitro. The levels of IL-2 (A), TNF-α (B), IFN-γ (C), IL-4 (D), IL-6 (E), IL-10 (F), and IL-17A (G) cytokines were detected by a Human Th1/Th2/Th17 Cytokine kit.

Round 2

Reviewer 3 Report

The authors have modified the manuscript sufficiently to address the concerns. I recommend publication of the manuscript.

Author Response

Dear reviewer,

Thank you very much for your endorsement of our revised manuscript. Your professional suggestions and comments have helped us to enhance the quality of our manuscript. Thanks again!

Best regards,

Wenping Gong, Ph.D.